# The Use of Platelet-Rich Fibrin in Combination with Synthetic Bone Grafting: A Systematic Review

**DOI:** 10.3390/biomedicines13092266

**Published:** 2025-09-15

**Authors:** Rosana Costa, Alicia Carvalho, Paula López-Jarana, Vitória Costa, Marta Relvas, Filomena Salazar, Tomás Infante da Câmara, Miguel Nunes Vasques, Marco Infante da Câmara

**Affiliations:** 1Department of Medicine and Oral Surgery, University Institute of Health Sciences (IUCS-CESPU), 4585-116 Gandra, Portugal; a29168@alunos.cespu.pt (A.C.); paula.jarana@iucs.cespu.pt (P.L.-J.); vitoria_gcosta@hotmail.com (V.C.); marta.relvas@iucs.cespu.pt (M.R.); filomena.salazar@iucs.cespu.pt (F.S.); up202106537@fmdup.up.pt (T.I.d.C.); miguel.vasques@cespu.pt (M.N.V.); marco.camara@iucs.cespu.pt (M.I.d.C.); 2Oral Pathology and Rehabilitation Research Unit (UNIPRO), University Institute of Health Sciences (IUCS-CESPU), 4585-116 Gandra, Portugal

**Keywords:** platelet-rich fibrin, sinus floor augmentation, bone substitutes, synthetic bone graft, synthetic hydroxyapatite, beta-tricalcium phosphate

## Abstract

**Background:** In atrophic posterior maxillary regions, sub-antral surgery is often used for rehabilitation with implants. In order to stimulate bone regeneration, autogenous, xenogenic, alloplastic and platelet-rich fibrin (PRF) grafts are commonly used. **Aim:** To assess the effectiveness of PRF alone or combination with synthetic bone substitutes on bone formation, implant stability, and survival in sub-antral surgery. **Materials and Methods:** A literature review was carried out from September 2024 to April 2025, according to PRISMA guidelines using the PubMed, Cochrane Library, Wiley, ScienceDirect, and Web of Science databases. From a total of 601 articles identified, 11 met the inclusion criteria and were selected for analysis. **Results:** PRF in combination with synthetic materials has shown potential benefits, especially in increasing biomechanical stability and bone formation. Although, most studies have not reported statistically significant differences when comparing the use of synthetic material alone against its combination with PRF. **Discussion:** The use of synthetic grafts in combination with PRF has become increasingly common in sub-antral implant procedures. PRF promotes angiogenesis, osteoprogenitor cell differentiation and bone regeneration, favouring the healing and remodelling process of the tissues, as well as greater stability and longevity of the implant. **Conclusions:** The combination of PRF with synthetic bone grafting shows promising results; however, further studies are needed to confirm the efficacy of PRF in maxillary sinus grafts in conjunction with the use of biomaterials.

## 1. Introduction

The use of dental implants has become a widely adopted practice in the rehabilitation of missing teeth, promoting not only functional restoration but also dental aesthetics. In cases of bone loss in the posterior maxillary region, subantral bone grafting is an essential procedure to increase vertical bone height, enabling the placement of implants with a high success rate—reaching over 90%—even in cases of severe bone resorption. However, one of the challenges following tooth extraction is the progressive loss of height and width of the alveolar bone in the posterior maxilla, which may compromise the success of implant placement [1,2]. The maxillary sinus lift procedure aims to increase bone volume in the post-anterior maxilla by elevating the sinus membrane and placing a bone graft, often supplemented with biomaterial. The amount of residual bone and the bucco-palatal distance determine the healing time of the maxillary sinus graft, which can vary between immediate implant placement and up to nine months of bone healing [3,4,5].

In 1976, Tatum developed the lateral window technique, which allowed access to the sinus floor via the lateral wall of the alveolus [6]. Later, in 1986, he introduced the transalveolar sinus lift technique for the first time, which was subsequently modified by Summers in 1994. This modification involved the use of a set of osteotomes of increasing diameter to condense the soft maxillary bone, creating an upward fracture in the sinus floor [7,8,9]. In 2013, Salah Huwais introduced a minimally traumatic osteotomy preparation technique known as osseodensification [10,11,12].

The maxillary sinus lift technique may or may not be performed using bone graft materials. Not using biomaterial requires sufficient residual vertical alveolar bone height in the posterior region of the maxilla [13,14]. The lack of sufficient bone height (<4 mm) in the edentulous maxilla below the maxillary sinus floor makes it necessary to use a bone graft. There are several bone graft materials that can be used to perform this procedure. These materials can vary in terms of their origin, such as autografts, allografts, xenografts, alloplastic grafts, and the use of platelet-rich fibrin (PRF) [15].

PRF stimulates cell migration, proliferation, and differentiation, as well as promoting the formation of new blood vessels. It has been widely used in regenerative treatments in the oral cavity [16,17,18]. Its use has shown potential for improving the osseointegration of dental implants and promoting better soft tissue healing after surgery, functioning as a biocompatible matrix that continuously releases growth factors. Thus, in addition to promoting osseointegration, PRF facilitates tissue regeneration and increases new bone formation [19,20,21].

PRF, introduced by Choukroun et al. [22] 2006, is a three-dimensional fibrin structure rich in platelets and cytokines located in the central layer between the red blood cells and plasma, which is obtained through a single centrifugation of non-anticoagulated blood without additives. This material is enriched with platelets and growth factors, which accelerate healing and promote bone formation and collagen matrix, making it a valuable biomaterial in bone regeneration procedures [23,24,25]. According to the literature, the addition of PRF to bone graft materials not only accelerates the healing process but also minimises bone loss, promoting better preservation of both the height and width of the alveolar ridge [1,21].

The main objective of this systematic review is to evaluate the use of autologous fibrin and/or synthetic bone in sub-antral surgery with regard to increased bone formation, stability, and implant survival.

## 2. Materials and Methods

This systematic review was conducted in accordance with the recommendations of the Preferred Reporting Items for Systematic Review and Meta-Analysis (PRISMA) statement [26]. The International Prospective of Systematic Reviews (PROSPERO) registered the study protocol for this systematic review under number CRD420251108710.

A search was conducted of articles available in the PubMed, Cochrane Library, Wiley Online Library, and Science Direct databases from September 2024 to April 2025.

The search strategy used was:

((((platelet rich fibrin[MeSH Terms]) OR (PRF)) AND ((augmentation, sinus floor[MeSH Terms]) OR (sinus lift[MeSH Terms]) OR (Lateral Sinus Floor Augmentation)) AND ((bone substitutes[MeSH Terms]) OR (synthetic bone graft) OR (alloplastic graft) OR (biomaterials[MeSH Terms]) OR (bone morphogenetic proteins[MeSH Terms]) OR (proteins[MeSH Terms]) OR (synthetic hydroxyapatite) OR (nanocrystalline[MeSH Terms]) OR (nanoporous) OR (actifuse synthetic bone graft[MeSH Terms]) OR (beta-tricalcium phosphate[MeSH Terms])))), for PubMed, Cochrane Library, and Wiley.

((((platelet rich fibrin[MeSH Terms]) OR (PRF)) AND ((augmentation, sinus floor[MeSH Terms]) OR (sinus lift[MeSH Terms]) OR (Lateral Sinus Floor Augmentation)) AND ((synthetic bone graft) OR (biomaterials[MeSH Terms])))), for Science Direct.

ALL = ((((platelet rich fibrin) OR (PRF)) AND ((augmentation, sinus floor) OR (sinus lift) OR (Lateral Sinus Floor Augmentation))) AND ((bone substitutes) OR (synthetic bone graft) OR (alloplastic graft) OR (biomaterials) OR (bone morphogenetic proteins) OR (proteins) OR (synthetic hydroxyapatite) OR (nanocrystalline) OR (nanoporous) OR (actifuse synthetic bone graft) OR (betatricalcium phosphate))), for Web of Science.

Data selection was performed by A.C. and systematically reviewed by R.C. to demonstrate intra- and inter-examiner reliability. A third reviewer (M.I.C.) reached agreement when the other reviewers disagreed.

A total of 601 articles were found on the subject. After applying the inclusion and exclusion criteria, 11 articles were selected. The article selection methodology is available in the flowchart shown in Figure 1. The eligibility criteria were organised using the PICOS (“Population”, “Intervention”, “Comparison”, “Outcomes”, and “Study design”) strategy (Table 1):

Using PICO analysis, the following research question is formulated: “Does the use of PRF and/or synthetic bone materials increase bone formation, stability, and implant survival after sub-antral surgery?”

Eligibility criteria were organised using the PICO methodology as follows:

The inclusion criteria were articles in English, Portuguese, or Spanish; studies conducted on humans; Research studies related to PRF and/or synthetic bone; randomised controlled clinical trials, cross-sectional studies, case-control studies, and cohort studies; and studies without research date restrictions. The exclusion criteria were studies based on the use of non-synthetic bone grafts such as autografts, allografts and xenografts in maxillary sinus elevation; studies not related to the topic; articles without abstracts; articles involving animal studies; systematic reviews; and meta-analyses.

### 2.1. Sample Data Extraction

The collected data were analysed using a results table based on the author, group, study objective, inclusion and exclusion criteria, sample size, number of sinus lift procedures, inherent complications, placement time after surgery, materials used, and results.

### 2.2. Quality Assessment and Risk of Bias

To evaluate the methodological quality and reliability of each study, the Joanna Briggs Institute (JBI) Risk of Bias Assessment Tool (2017 version) was used. Different checklists were applied depending on the study type—randomised controlled trials, cohort studies, or case series—with responses categorised as: Yes (Y), No (N), Unclear (U), or Not Applicable (NA).

## 3. Results

In total, 601 articles were initially identified. After excluding duplicates and reading the title and abstract, the remaining articles were completely read. In the end, 11 articles were included.

Figure 1 shows the detailed selection strategy for the articles.

### 3.1. Sample Characteristics for Study Quality

The methodological quality of the included studies was assessed using the Joanna Briggs Institute checklists, applied according to the type of study, namely randomised controlled trials, cohort studies, and case series studies.

In randomised clinical trials, four presented a low risk of bias—Belouka et al. [27], 2016, Cömert Kılıç et al. [28] 2017, Amam et al. [29], 2023, and Anis et al. [30], 2024—while Angelo et al. [31], 2015 was considered to have moderate risk. Regarding cohort studies, Wolf et al. [32], 2014 and Ghanaati et al. [33], 2014 were considered low risk, while Bosshardt et al. [34] 2014 and El Hage et al. [35], 2012 presented moderate risk. The two case series studies included Francisco et al. [36], 2024 and Canullo et al. [37], 2012, and revealed a low risk of bias.

Thus, overall, the evidence included in this review is considered to have a low to moderate risk of bias and is methodologically adequate to support the conclusions, although with some limitations to consider.

The quality assessments of the studies are presented in Table 2 for randomised controlled clinical trials, in Table 3 for cohort studies, and in Table 4 for case series studies.

### 3.2. Characteristics of the Included Studies

For each eligible study included in this systematic review, data was collected on general characteristics such as author, group, study objective, inclusion and exclusion criteria, sample, and number of sinus lift procedures. The inherent complications, the time of placement after surgery, the materials used and the results were also analysed, as shown in Table 5.

### 3.3. Study Designs

Regarding the type of study of the articles analysed, five are randomised clinical trials (46.0%), four are cohort studies (36.0%), and two are case series studies (18.0%).

Figure 2 shows the study designs of the articles included in this systematic review.

## 4. Discussion

Eleven studies addressing different bone regeneration strategies in sub-antral elevation or maxillary sinus elevation procedures were analysed. The study by Francisco et al. [36] 2024 evaluated the use of autologous platelet concentrates, namely PRF, in combination with NanoBone^®^, a synthetic bone graft material. Other studies like Cömert Kılıç et al. [28] 2017 analysed the use of PRF combined with β-TCP in sinus lift surgeries. The studies of El Hage et al. [35] 2012, Canullo et al. [37] 2012, Wolf et al. [32] 2014, Ghanaati et al. [33] 2014, Bosshardt et al. [34] 2014, and Belouka et al. [27] 2016 focused on the use of synthetic bone substitutes, more specifically nanocrystalline hydroxyapatite. Angelo et al. [31] 2015 analysed self-hardening mouldable synthetic biomaterials associated with fibrin. Lastly, Amam et al. [29] 2023 and Anis et al. [30] 2024 compared different types of synthetic grafts, such as HA, β-TCP, and CS, with and without the use of PRF.

### 4.1. Subantral Surgery

Sub-antral surgery is a widely used technique in rehabilitation with implants in posterior areas of the maxilla, where bone height is often insufficient due to alveolar bone resorption and maxillary sinus pneumatisation after tooth loss [35,36]. This procedure creates a space between the sinus membrane and the sinus floor, which allows new bone formation in order to allow rehabilitation with implants [27]. There are two main approaches: the lateral approach, which is more invasive, and the transcrestal approach, which is less traumatic and generally associated with a simpler recovery [31,38]. Traditionally, this space can be filled with bone graft materials, such as autogenous, allogeneic, xenogeneic, or synthetic bone. However, more recently, some studies have explored the use of platelet-rich fibrin (PRF) as a filling material, either alone or in combination with other graft materials [28,31,36,38].

### 4.2. Synthetic Bone Substitutes in Sub-Antral Surgery

Bone regeneration in this surgical context has been the subject of several studies, which analyse the effectiveness of different biomaterials in new bone formation and the osseointegration of implants. Among these, fully synthetic materials such as β-tricalcium phosphate, calcium sulphate, and nanocrystalline hydroxyapatite stand out, characterised by their biocompatibility, osteoconductive properties, and gradual resorption capacity. These materials act as a support and are beneficial for bone augmentation without inducing adverse inflammatory responses. Several authors have demonstrated that β-TCP promotes effective and predictable bone regeneration, with new bone formation and balanced degradation of the biomaterial. Similarly, studies with NanoBone^®^ have demonstrated 24.0% to 35.0% of new vital bone in 6-month periods, with bone–implant contact already occurring at 3 months, a low resorption rate, and good tissue and vascular response, confirming the efficacy and safety of these substitutes in sub-antral surgeries [27,28,32,33,34,35,37,39].

Contrasting with the opinions of other authors available, the systematic review conducted by Ortega-Mejia et al. [40] 2020 introduces uncertainty by concluding that there is no robust evidence confirming the benefits of using synthetic materials exclusively in maxillary sinus procedures. The authors analysed several studies and found no significant differences in bone formation, increased bone height, or implant stability when PRF was used alone. In addition, they report that the combination of PRF with other biomaterials did not show significant advantages over the isolated use of these biomaterials [40].

### 4.3. PRF and Its Application in Sub-Antral Surgeries

PRF is an autologous biomaterial obtained by centrifuging the patient’s own blood, which produces a fibrin matrix rich in platelets, leukocytes, and growth factors. This matrix acts as a biological reservoir that promotes angiogenesis, osteoprogenitor cell differentiation and bone regeneration, supporting the healing process and tissue remodelling [30,36].

Its application in oral surgery, particularly in sub-antral surgeries or what is commonly referred to as maxillary sinus elevation, has been mostly studied in association with synthetic biomaterials. The results are divergent, Cömert Kılıç et al. [28] 2017 evaluated the combination of PRF with β-TCP and concluded that the bone formation values were like those obtained with β-TCP alone, with no statistically significant differences observed between the groups. On the other hand, Francisco et al. [36] 2024 found that in a control group (with NanoBone^®^), the percentage of new bone was 19.5 ± 3.0%, inert particles were 23.4 ± 5.7% and connective tissue was 57.0 ± 3.5%; in the test group (NanoBone^®^ + PRF), 27.5 ± 4.9% of new bone was formed, 23.0 ± 3.7% was inert particles and 49.4 ± 2.8% was connective tissue, suggesting a positive influence of the addition of PRF on the quality and maturation of the formed bone tissue [28,36].

PRF has proven to be a promising biological adjuvant, contributing to more balanced bone regeneration, namely by promoting vascularisation and through its anti-inflammatory action. Its effectiveness, however, seems to depend on several factors, such as the type of biomaterial with which it is combined, the technique used, and the clinical characteristics of the patient [30,36].

### 4.4. Synthetic Bone Versus PRF

The comparison between synthetic biomaterials and PRF in bone regeneration in maxillary sinus lift surgery reveals significant differences in their functions and results. Synthetic grafts, such as β-tricalcium phosphate, calcium sulphate, and nanocrystalline hydroxyapatite act as osteoconductive materials, providing structure and volume to bone defects, and serve as a matrix for new bone regeneration [26,32,35,37].

Beta-Tricalcium phosphate (β-TCP) is considered a recommended bone substitute for maxillary sinus elevation, as it stimulates bone formation due to its absorbent nature and volume stability [41,42,43].

In comparison with other biomaterials currently available, calcium sulphate (CS) has a longer history of clinical use and is widely recognised as a well-tolerated substance with applications in bone regeneration, giving it a special place in the field of regenerative materials. This material is almost completely absorbed in vivo without causing a significant inflammatory response, which is essential in this surgical procedure [44].

The literature shows that these materials ensure good dimensional stability and tissue integration, with bone formation percentages between 25.0% and 35.0% in periods of 6 to 12 months and are considered safe and effective as bone substitutes in sub-antral procedures [26,32,35,37].

PRF clinical effects are caused by growth factors and cytokines released during PRF production, including transforming growth factor-β1, insulin-like growth factor-1, vascular endothelial growth factor, platelet-derived growth factor α and β, interleukin 1β, interleukin 4, and tumour necrosis factor α. By promoting collagen synthesis, which strengthens wounds and causes callus formation, these components enhance both soft and hard tissue healing [45].

PRF actively contributes to the modulation of the regenerative environment, promoting angiogenesis, cell differentiation, and tissue reorganisation. Its use has shown benefits, especially when combined with synthetic biomaterials, improving the quality of the bone formed and the vascularisation of the graft, although the gains in bone volume are not significantly higher than those obtained through the isolated use of the materials [28,36].

Thus, while synthetic grafts provide essential structural support for bone regeneration, PRF enhances the biological response to the graft. The combination of both, although it does not significantly increase the amount of bone formed, tends to improve its quality and maturation, and may represent an advantageous complementary improvement in specific clinical contexts [30,36].

### 4.5. Complications and Perforation of Schneider’s Membrane

Schneider’s membrane is a thin mucous lining of the maxillary sinus, the integrity of which is essential during sinus lift surgery. Its perforation can compromise graft stability, promote biomaterial migration, and increase the risk of infection. Among the studies, perforations were only observed in the study by Belouka et al. [27] 2016, in 3 of the 44 cases (6.8%), all of which were resolved with the application of resorbable collagen membranes, without the need for further treatment.

In the study by El Hage et al. [35] 2012, although no perforations occurred, two postoperative infections were recorded. One of the cases presented persistent oedema and significant pain 7 weeks after surgery, requiring reoperation. The other was treated with antibiotic therapy and local irrigation, without further complications.

The presence of pain and oedema is also addressed in other studies. Francisco et al. [36] 2024 specifically mention that none of the six patients showed signs of infection, severe pain, or significant oedema in the immediate or late postoperative period. The use of PRF in this case may have contributed to a more controlled inflammatory response, given its modulating action on tissue healing and regeneration [36].

In the remaining studies that used synthetic bone in isolation in one of the groups, such as Bosshardt et al. [34] 2014, or in combination with PRF, as in the study by Francisco et al. [36] 2024, no membrane perforations or complications such as prolonged pain or oedema were reported. In Canullo et al. [37] 2012, even with the placement of mini-implants in regions with low bone height, no significant adverse complications were reported, which reinforces the safety of the materials used. This can be explained by their high biocompatibility, such as that of nanocrystalline hydroxyapatite, which has a porosity and structure like natural bone, favouring tissue integration without triggering a significant inflammatory response. In addition, the combination with PRF, which is rich in anti-inflammatory and angiogenesis-promoting factors, may have contributed to reducing postoperative discomfort and accelerating healing [30,34,36,37].

Overall, the data indicate a low incidence of complications. Pain and swelling, when present, were limited and easily controlled, especially in cases where PRF was used, suggesting that it may play a beneficial role in reducing inflammation and improving post-operative comfort.

### 4.6. Histology, Bone Regeneration, and New Bone Formation

Bone regeneration after or during maxillary sinus elevation has been analysed by several authors using histological analyses to evaluate parameters such as new bone formation, the percentage of remaining biomaterial and the presence of soft tissue. Studies like Bosshardt et al. [34] 2014, Canullo et al. [37] 2012, and Francisco et al. [36] 2024 consistently demonstrated active and well-organised bone regeneration, confirming the effectiveness of grafting procedures in the maxillary sinus.

The influence of platelet-rich fibrin (PRF) on bone regeneration is still controversial; Cömert Kiliç et al. [28] 2017 and Bosshardt et al. [34] 2014 suggest that PRF did not promote statistically significant differences in bone formation when compared to β-TCP alone or collagen membrane. In the same way, Cömert Kiliç et al. [28] 2017, in a previous study, evaluated the role of PRP instead of PRF, and it was also concluded that the addition of PRP or PRF to β-TCP did not bring significant benefits in bone regeneration [28,34,46].

In several articles, areas of new bone with the presence of osteoblasts, well-defined or forming Haversian canals, and lamellar structures were observed, evidencing progressive bone maturation [32,34,36,37]. Francisco et al. [36] 2024 in particular showed the presence of thick trabeculae of lamellar bone and vascularised connective tissue with fibroblasts, as well as active osteoblasts and osteoclasts associated with graft particle resorption, especially in the group treated with PRF. Similarly, Canullo et al. [37] 2012 documented the formation of mineralised lamellar bone with well-established and delimited Haversian systems around the graft particles, and direct bone-implant contact without soft tissue interposition.

Corroborating this, Bosshardt et al. [34] 2014, reported similar levels of newly formed bone, with evident integration of the biomaterial and trabecular organisation. Wolf et al. [32] 2014, also confirmed the formation of new bone at different stages of maturation, with no significant difference between groups, highlighting the biocompatibility of NanoBone^®^. This evidence suggests that the biomaterials used, especially when combined with PRF, promote a stable environment with a histological structure compatible with favourable regeneration, although some authors conclude that the differences are not significant [32].

#### Quantitative Analysis: Bone Formed, Residual Biomaterial, and Soft Tissue

The quantitative data presented in the studies analysed reveal a consistent trend in new bone formation, with variations that seem to imply both the type of biomaterial used and the possible association with PRF. Francisco et al. [36] 2024 revealed an average of 27.5 ± 4.9% vital bone in the PRF-treated group, which was higher than that observed in the control group (19.5 ± 3.0%), where only NanoBone^®^ was used. The percentages of remaining biomaterial were similar between both groups (~23%), but there was a lower proportion of connective tissue in the experimental group, which may reflect a more advanced state of maturation.

In the same way, Bosshardt et al. [34] 2014 observed levels of newly formed bone between 28.6% and 28.7%, with a balanced distribution between residual biomaterial (25.0–26.0%) and connective tissue (approximately 45.0%), suggesting effective graft integration and viable bone regeneration. Canullo et al. [37] 2012, observed particularly promising results just three months after surgery, with an average of 32.5% ± 5.0% total bone, of which 20.6% ± 2.96% corresponded to newly formed bone and 11.87% ± 3.27% to native bone. A proportion of 38.26% ± 8.07% of residual biomaterial and 29.23% ± 5.18% of medullary tissue was also observed. This leads us to conclude that these results are particularly relevant, given the short healing interval and the anatomical limitations involved.

In the case of Wolf et al. [32] 2014, they evaluated and analysed the effect of age on regeneration after 7 months. In the 66–71 age group, 20.57% ± 6.95% new bone, 39.28% ± 11.58% residual NanoBone^®^, and 40.15% ± 10.93% connective tissue were observed; in the younger group, aged 41–52 years, the values were similar: 22.27% ± 4.31% new bone, 39.13% ± 8.86% biomaterial, and 38.59% ± 9.97% connective tissue. According to these results, no statistically significant differences were found between age groups, reinforcing the consistency of the biological response to NanoBone^®^ regardless of patient age.

### 4.7. Survival Rate and Stability

The stability and survival of implants placed after maxillary sinus elevation are debated in several studies, with contrasting opinions regarding the performance of the different biomaterials used.

Angelo et al. [31] 2015 observed that mouldable and self-hardening biomaterials (Easy-Graft^®^ CRYSTAL and CLASSIC) resulted in significantly higher insertion torque (IT) values (up to 52.5 Ncm). For example, the group treated with Easy-Graft^®^ CLASSIC + PRF had a mean IT of 46.89 Ncm, higher than the same biomaterial without PRF (42.51 Ncm) and significantly higher than native bone (27.87 Ncm). This result suggests that PRF increases the density and biomechanical stability of the graft, probably by accelerating vascularisation and cell recruitment in the grafted area, which suggests high primary stability of the implants compared to autologous bone, especially when associated with PRF [31].

From another perspective, Ghanaati et al. [33] 2014 observed a survival rate of 95.8% after 3 years, with average Periotest^®^ values of −2.74, indicating excellent implant stability. Interestingly, they concluded that reducing the graft healing time from 6 to 3 months did not compromise implant stability, pointing to the effectiveness of NanoBone^®^ as a reliable biomaterial in the short and long term.

Similarly, El Hage et al. [35] 2012 reported a survival rate of 94.74% one year after implant placement in maxillary sinuses previously grafted with NanoBone^®^. The implants were placed, on average, 14 months after grafting, and stability was clinically confirmed with prosthetic rehabilitation in 18 of the 19 implants placed, reinforcing the predictability of this biomaterial even in long-term healing protocols.

Although Francisco et al. [36] 2024 did not provide IT or RFA values, the authors performed a histological analysis on maxillary sinuses grafted with NanoBone^®^ and PRF and found an increase in the amount of vital bone formed compared to the group with NanoBone^®^ alone. This increase in bone regeneration suggests that the use of PRF may improve the quality and stability of the bone bed for subsequent implant placement [36].

In the remaining studies analysed, although no specific stability values such as IT or RFA were specified, all implants were successfully osseointegrated after regeneration with synthetic materials, often combined with PRF. For example, the article by Francisco et al. [36] 2024, which revealed, through histological analysis after 6 months, effective bone regeneration results, with higher percentages in the PRF group, allowing for the subsequent successful placement of implants in all cases evaluated.

In summary, the authors corroborate the idea that implant placement after sub-antral regeneration with synthetic biomaterials, with or without PRF, is clinically feasible and has high survival rates [10,31].

### 4.8. Study Limitations

Despite the relevance of the results analysed, this topic has some important limitations. One of the main limitations relates to the methodology of the articles included, which vary in terms of study type, time, methods of assessing bone regeneration, and sample size, which was reduced in some studies, limiting the generalisation of the results [32,34,36,37].

In addition, the diversity of materials used makes direct comparison between studies difficult. Different types of synthetic bone were included, such as β-TCP, calcium sulphate, and nanocrystalline hydroxyapatite, as well as different forms of PRF, such as L-PRF and A-PRF, with different preparation and application protocols. This methodological variability may influence both histological and quantitative results [31,33,38]. Another relevant limitation is the lack of consistent data on postoperative complications. Some articles report no pain, oedema, or perforation of Schneider’s membrane, but often do not specify whether these parameters were systematically evaluated, which makes it difficult to draw conclusions about the safety of the materials [34,36].

Finally, the lack of studies with recurrent follow-up compromises the assessment of graft stability and long-term implant success. Most studies focus on healing periods of 3 to 9 months, and further research is needed with randomised clinical trials, larger samples, and standardised protocols to increase the robustness of the scientific evidence.

## 5. Conclusions

We can conclude that the studies analysed demonstrate that PRF has significant potential in bone regeneration in sub-antral surgery. PRF acts as a bioactive matrix, promoting angiogenesis and cell differentiation, and the results of some studies suggest that it can accelerate healing and can be used alone or in combination with synthetic biomaterials. Despite its promising properties, the results are not unanimous: some studies have not observed statistically significant differences when compared to synthetic materials used alone.

At the same time, synthetic bone substitutes, such as nanocrystalline hydroxyapatite, β-TCP, and CS, have proven to be safe and effective materials in sub-antral bone regeneration, providing a stable implant bed. These biomaterials have good biocompatibility, gradual remodelling, and satisfactory bone integration, with no reports of relevant complications.

The combination of PRF with synthetic biomaterials has shown, in some studies, positive effects on healing and primary stability of implants, increased vascularisation and higher insertion torque values.

Thus, PRF proves to be particularly useful in the initial phase of regeneration, contributing to better tissue healing and ensuring biological conditions conducive to bone regeneration. However, further studies with larger samples and long-term follow-up are needed to clarify the real clinical impact of its application in combination with synthetic bone substitutes.

## Figures and Tables

**Figure 1 biomedicines-13-02266-f001:**
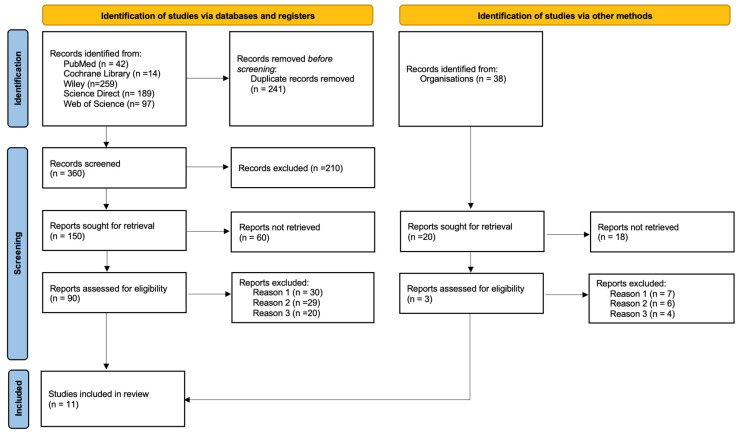
PRISMA Flow diagram. Reason 1: Irrelevant to the topic; Reason 2: After reading the abstract; Reason 3: the population included patients who underwent maxillary sinus lift without synthetic bone.

**Figure 2 biomedicines-13-02266-f002:**
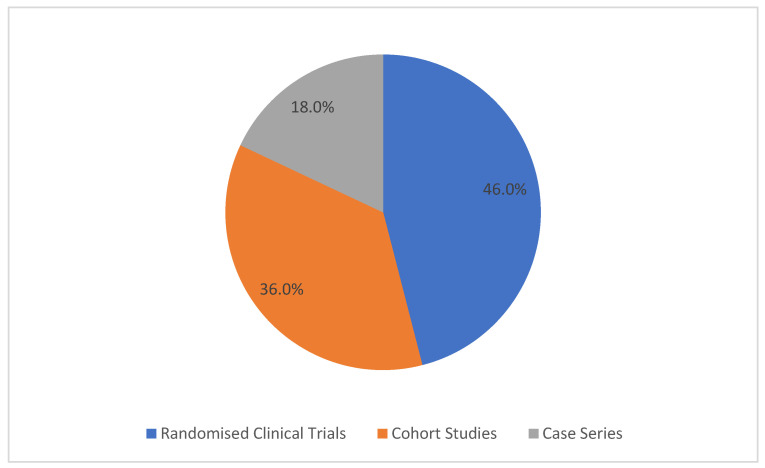
Study designs of the included studies.

**Table 1 biomedicines-13-02266-t001:** PICOS Strategy.

P	Population	Patients requiring sub-antral bone grafting
I	Intervention	Use of synthetic bone
C	Comparison	Use of synthetic bone with PRF
O	Outcomes	Analysing the PRF technique in sub-antral surgery

**Table 2 biomedicines-13-02266-t002:** Joanna Briggs Institute Critical Appraisal Checklist for randomised controlled clinical trial.

	Was the Randomisation Method Adequate?	Was the Allocation Method Adequate?	Were the Groups Similar at the Start of the Study?	Were the Participants Blinded?	Were the Professionals Who Administered the Interventions Blinded?	Were the Outcome Assessors Blinded?	Were the Interventions Clearly Described and Applied Equally in Both Groups?	Was the Primary Outcome Clearly Defined and Measured?	Was There an Intention-to-Treat Analysis?	Were Losses and Exclusions Described?	Were Complications or Adverse Events Reported?	Were the Study Results Accurate and Reliable?	Were the Study Results Relevant to Clinical Practice?
Angelo et al. [31], 2015	Y	Y	Y	U	U	U	Y	Y	N	NA	N	Y	Y
Belouka et al. [27] 2016	Y	Y	Y	U	U	U	Y	Y	Y	Y	Y	Y	Y
Cömert Kılıç et al. [28], 2017	Y	Y	Y	N	N	Y	Y	Y	I	S	S	S	S
Amam et al. [29], 2023	Y	Y	Y	N	N	N	Y	Y	Y	Y	Y	Y	Y
Anis et al. [30], 2024	Y	Y	Y	N	N	Y	Y	Y	Y	Y	Y	Y	Y

(Y)—Yes, (N)—No, (U)—Unclear, (NA)—Not applicable.

**Table 3 biomedicines-13-02266-t003:** Joanna Briggs Institute Critical Appraisal Checklist for cohort studies.

	Were the Two Groups Similar and Recruited from the Same Population?	Were Exposures Measured in a Similar Way to Assign People to Exposed and Unexposed Groups?	Was Exposure Measured in a Valid and Reliable Way?	Were Confounding Factors Identified?	Were Strategies Defined to Deal with Confounding Factors?	Were the Groups/Participants Free of the Outcome at the Start of the Study (or at the Time of Exposure)?	Were the Outcomes Measured in a Valid and Reliable Manner?	Was the Follow-Up Time Reported and Sufficient for the Outcomes to Occur?	Was the Follow-Up Complete, and If Not, Were the Reasons for Follow-Up Losses Described and Analysed?	Were Strategies Used to Deal with Incomplete Follow-Up Losses?	Was an Appropriate Statistical Analysis Used?
Dieter D. Bosshardt et al. [34], 2014	Y	Y	Y	N	N	Y	Y	Y	U	NA	Y
Wolf et al. [32] 2014	Y	Y	Y	U	N	Y	Y	Y	Y	NA	Y
El Hage et al. [35], 2012	NA	NA	Y	N	N	Y	Y	Y	Y	N	N
Ghanaati et al. [33], 2014	Y	Y	Y	N	N	Y	Y	Y	Y	N	Y

(Y)—Yes, (N)—No, (U)—Unclear, (NA)—Not applicable.

**Table 4 biomedicines-13-02266-t004:** Joanna Briggs Institute Critical Appraisal Checklist for Series cases.

	Were the Inclusion Criteria Well Defined?	Was the Condition Measured Reliably?	Were Valid Methods Used for the Condition of All Participants Included?	Did the Case Series Have Consecutive Inclusion of Participants?	Did the Case Series Include All Eligible Participants?	Was There a Clear Description of the Demographics of the Study Participants?	Was There a Clear Description of the Clinical Information of the Participants?	Were the Results or Follow-Up of the Cases Clearly Described?	Was There a Clear Description of the Demographic Information of the Site(s)/Clinic(s) Where the Study Was Conducted?	Was the Statistical Analysis Appropriate?
Francisco et al. [36], 2024	Y	Y	Y	U	Y	Y	Y	Y	N	Y
Canullo et al. [37], 2012	Y	Y	Y	N	Y	N	Y	Y	N	Y

(Y)—Yes, (N)—No, (U)—Unclear.

**Table 5 biomedicines-13-02266-t005:** Descriptive analysis of selected scientific articles.

Author	Type of Study	Study Group	Inclusion Criteria	Exclusion Criteria	Objective	Sample	Sinuses	Complications	Implant Placement	Outcome Measurements	Results/Conclusion
El Hage et al. [35], (2012)	Prospective cohort study	Human	Partially edentulous patients with posterior maxillary bone defects; residual bone height ≤ 3 mm	Severe systemic conditions	To evaluate the percentage of vertical resorption of NanoBone^®^ grafts following subantral surgery, and the implant survival rate after 1 year	N = 8 Single group (five female, three male)—NanoBone^®^ + autologous blood + collagen membrane Mean age: 53 years	11	Two postoperative infections; one implant loss	12 months	Vertical graft resorption; Implant survival rate	Mean graft resorption: 8.84% ± 5.32% Implant survival rate after 1 year: 94.74% 18 implants successfully osseointegrated and rehabilitated; NanoBone^®^ graft showed good dimensional stability
Canullo et al. [37], (2012)	Case series	Human	Need for rehabilitation in the posterior maxilla with residual bone height of 1–2 mm	Chronic sinusitis Acute infections Respiratory allergies Use of bisphosphonates	To histologically evaluate bone regeneration with NanoBone^®^ and BIC after 3 months	N = 10 patients Single group—NanoBone^®^ Mean age: 54 years	10	N/R	Mini-implant placed at the time of surgery to maintain space	% New Bone; Residual graft material; Bone marrow content and BIC.	New bone: 20.64% ± 2.96% Residual NanoBone^®^: 38.26% ± 8.07% Bone marrow: 29.23% ± 5.18% BIC: 26.02% ± 5.46% No connective tissue observed at the implant surface.
Ghanaati et al. [33], (2014)	Prospective cohort study	Human	Edentulous patients in the upper molar region; Severely resorbed maxilla; Age between 34 and 77 years	Chronic infections in the maxillary region Uncontrolled diabetes and other systemic conditions Use of bisphosphonates	Evaluate the impact of NanoBone^®^ synthetic bone integration time on implant stability at 3 and 6 months after sinus lift surgery	N = 14 Group 1: 3 months: NanoBone^®^, sevent patients (four men, three women) Mean age: 53 years Group 2: 6 months: NanoBone^®^, seven patients (three men, four women) Mean age: 53 years	14	One implant lost in the test group	3 or 6 months	% New Bone; Implant survival at 3 years; Periotest; Presence of osteolysis; BOP; PB; REC	Group 1: New bone: 24.89% ± 10.22% Implant survival: 94.1% Mean Periotest: 2.94 Group 2: New bone: 31.29% ± 2.29% Implant survival: 100% Mean Periotest: 2.29 No osteolysis or mobility Three months after the surgical procedure, it is already possible to achieve a stable and lasting restoration, retained by the implant, which can contribute to a reduction in healing time.
Dieter D. Bosshardt et al. [34], (2014)	Cohort study—Histological and histomorphometry analysis	Human	Vertical height of the edentulous maxillary ridge < 4 mm Age between 41 and 64 years Patients referred to the Department of Stomatology and Oral Surgery at the University of Geneva	Smokers; Acute or chronic sinus disease	Evaluate bone regeneration following maxillary sinus lift using nanocrystalline hydroxyapatite in a silica gel matrix	N = 8 (seven female, one male) Group 1: NanoBone^®^ + collagen membrane (Bio-Gide^®^) three patients Age range: 41–64 years Group 2: NanoBone^®^ + PRF membrane five patients Age range: 41–64 years	16	N/R	7–11 months	Histomorphometry % of new bone; % of residual NanoBone^®^; % of soft tissue; Bone-material integration; TRAP+ cells and vascularization	Group 1: NanoBone^®^ + Bio-Gide^®^ New bone: 28.7% ± 5.4% Residual material: 25.5% ± 7.6% Soft tissue: 45.8% ± 3.2% Group 2: NanoBone^®^ + PRF membrane New bone: 28.6% ± 6.9% Residual material: 25.7% ± 8.8% Soft tissue: 45.7% ± 9.3% No significant difference compared to collagen group Similar osteointegration and histological pattern New bone formation following the use of nanocrystalline hydroxyapatite for maxillary sinus floor elevation in humans is comparable to values reported in other synthetic or xenogeneic bone substitute materials.
Wolf et al. [32], (2014)	Cohort study	Human	Need for maxillary sinus lift prior to implant placement; Residual subantral bone height between 3 mm and 7 mm	Severe systemic diseases Uncontrolled diabetes History of radiotherapy to the head and neck region Chemotherapy Sinus disease Active periodontal disease Smoking	Evaluate whether patient age influences bone formation, biomaterial resorption, and soft tissue development after sinus lift using NanoBone^®^	N = 17 patients (nine male, eight female) Group 1: NanoBone^®^: eight patients aged 66–71 years Group 2—NanoBone^®^: nine patients age: 41–52 years	20	N/R	7 months	% New Bone; % Residual NanoBone^®^; % Soft tissue; Presence of TRAP+ cells	Group 1—New bone: 20.57% ± 6.95%; Residual NanoBone^®^: 39.28% ± 11.58%; Soft tissue: 40.15% ± 10.93%; Group 2—New bone: 22.27% ± 4.31%; Residual NanoBone^®^: 39.13% ± 8.86%; Soft tissue: 38.59% ± 9.97%; TRAP+ cells were present in both groups, with no significant differences observed.
Angelo et al. [31], (2015)	Randomised	Human	Anterior maxillary alveolar ridge with width < 3 mm and height > 14 mm	Platelet disorders; chronic sinusitis; infectious/metabolic diseases; chemotherapy/radiotherapy; use of antibiotics or anti-inflammatory drugs.	Evaluate maxillary bone regeneration and biomechanical implant stability using self-hardening synthetic biomaterials (SHB), either combined or not with PRF.	N = 82 Control Group: native bone Group 1: HA + β-TCP (Easy-Graft^®^ CRYSTAL) Group 2: β-TCP (Easy-Graft^®^ CLASSIC) Group 3: β-TCP + PRF (Easy-Graft^®^ CLASSIC + PRF) Age range: N/R	82	N/R	8 months	Insertion torque (IT) and standard deviation	Control Group (native bone): Mean IT: 31.1 ± 7.3 Ncm. Grupo 1: Mean IT: 43.2 ± 7.6 Ncm; +38.6% compared to control group; Grupo 2: mean IT: 39.4 ± 8.9 Ncm; +26.7% compared to control group Grupo 3: Mean IT: 41.2 ± 5.4 Ncm; +32.5% compared to control group; The use of CS, alone or in combination with PRF, was advantageous for obtaining repaired alveolar bone with improved (bio)mechanical stability.
Belouka et al. [27], (2016)	Randomised	Human	Age ≥ 18 years; Need for maxillary sinus elevation for rehabilitation with implants; At least two adjacent teeth missing in the posterior maxilla.	Age < 18 years; Uncontrolled systemic diseases (ASA > 2); Drug use and alcoholism; Active periodontal disease or chronic sinusitis. Patients who refused the use of synthetic grafts.	Histomorphometric comparison of bone regeneration in maxillary sinus elevation between two types of synthetic bone: nanocrystalline hydroxyapatite (Ostim^®^) vs. nanoporous hydroxyapatite (NanoBone^®^)	N = 44 Group 1—Nanocrystalline HA (Ostim^®^) 22 patients (9 women, 13 men) Age: average: 63 years Group 2—Nanoporous HA (NanoBone^®^) 22 patients (13 women, 9 men) Average age: 63 years	88	Ostim^®^—one implant lost	Immediate	% New Bone; % of remaining biomaterial; % of soft tissue; Histology and histomorphometry at 6 months.	% New bone: NanoBone^®^: 34.6% ± 9.2% Ostim^®^: 31.8% ± 11.6% *p* = 0.465 % Remaining biomaterial: NanoBone^®^: 30.0% ± 13.0% Ostim^®^: 28.4% ± 18.6% *p* = 0.828 % Soft tissue NanoBone^®^: 35.4% ± 6.8% Ostim^®^: 39.9% ± 11.1% *p* = 0.159 Both synthetic bone substitute materials were found to support bone formation in sinus floor elevation by osteoconductivity.
Cömert Kılıç et al. [28], (2017)	Randomised	Human	Adults with atrophic maxilla; Residual bone crest height ≤ 7 mm	Infections in the maxillary sinus; Haematological, neurological, or systemic diseases; Radiotherapy/chemotherapy; Inflammatory or malignant diseases in the head/neck region	Compare histological and histomorphometric results of surgery with β-TCP alone, β-TCP + P-PRP and β-TCP + PRF	N = 26 Group 1: β-TCP + P-PRP nine patients (four women, five men) Age: 22–51 years Group 2: β-TCP + PRF eight patients (three women, five men) Age: 22–51 years Group 3-Control: β-TCP nine patients (two women, seven men) Age: 22–51 years	26	Five perforations: two in the control group; one in the P-PRP group; two in the PRF group	6 months	% New Bone; % residual biomaterial; % soft tissue; Osteoblastic activity.	% of new bone: β-TCP + PRF: 35.2% ± 7.6% β-TCP + PRP: 30.4% ± 8.1% β-TCP: 27.3% ± 6.8% % of remaining biomaterial: β-TCP + PRF: 27.9% ± 7.4% β-TCP + PRP: 31.4% ± 6.3% β-TCP: 34.3% ± 5.9% % of soft tissue: β-TCP + PRF: 36.9% ± 6.3% β-TCP + PRP: 38.2% ± 5.7% β-TCP: 38.4% ± 5.4%. Intense osteoblastic activity, particularly in the PRF group.
Amam et al. [29], (2023)	Randomised	Human	Bilateral maxillary edentulism; Age between 45 and 70 years; Residual bone height between 0.5 and 5 mm	Metabolic diseases; Use of corticosteroids; Autoimmune, cardiovascular diseases, diabetes; Coagulation disorders.	Comparing CS and β-TCP grafts in maxillary sinus elevation	N = 9 Test group: CS + A-PRF Age: 45–70 Control group: β-TCP + A-PRF Age: 45–70	18	N/R	6 months	Vertical bone augmentation; Assessment by CBCT in the following phases: T0: preoperative; T1: immediate postoperative; T2: 6 months postoperative	Test group: CS + A-PRF Bone gain: 7.96 ± 2.78 mm (+372.8%) Control group: β-TCP + A-PRF Bone gain: 7.53 ± 1.15 mm (+353.2%) *p* > 0.05 in all comparisons T0: Reduced initial bone height; T1: Average increase in bone height: CS/A-PRF: +10.3 mm; β-TCP/A-PRF: +10.4 mm; T2: Reduction in bone height: CS/A-PRF: −2.35 mm; β-TCP/A-PRF: −2.79 mm. The use of CS or TCP combined with A-PRF proved to be advantageous and safe, with sufficient bone available for dental implant placement.
Francisco et al. [36], 2024	Case Series	Human	≥18 years; Posterior maxillary edentulism; Residual bone height ≤ 5 mm	Alcoholism and smoking; Diabetes; Heart disease; Use of bisphosphonates; Previous sinus pathology.	To evaluate PRF combined with NanoBone^®^ in bone regeneration of maxillary sinus elevation.	N = 6 (three male/three female) Test Group: NanoBone^®^: + PRF Control Group: NanoBone^®^: Age: N/R	12	N/R	6 months	% New Bone; % of inert particles; % Connective tissue; Histomorphometry at 6 months.	Test group—PRF + NanoBone^®^: New bone: 27.5% ± 4.9% Inert particles: 23.0% ± 3.7% Connective tissue: 49.4% ± 2.8% Control group—NanoBone^®^: -New bone: 19.5% ± 3.0% -Inert particles: 23.4% ± 5.7% Connective tissue: 57.0% ± 3.5% Mixing liquid PRF with NanoBone^®^ appears to slightly increase the amount of new bone formation and revascularization in comparison to using NanoBone^®^ alone.
Anis et al. [30], (2024)	Randomised	Human	Age > 18 years; Edentulous maxillary crest with width < 6 mm; Seibert class I defects.	Smoking; Systemic diseases; Pregnancy; Chemotherapy and radiotherapy.	Evaluating bone changes in maxillary ridges in split-crest surgeries with PRF versus PRF + NanoBone^®^	N = 40 Test group—20 Patients (13 women, 7 men) PRF + NanoBone^®^: Average age: 35 years Control group—PRF: 20 patients (11 women, 9 men) Average age: 35 years	N/R	Healing screw exposure in one case	Immediately	Crestal bone changes; Gain in horizontal bone width; Post-operative pain and oedema.	Group 1: PRF + NanoBone^®^: Vestibular bone resorption: 1.14 ± 0.63 mm; Lingual bone resorption: 1.47 ± 0.68 mm; Horizontal bone gain: 1.29 ± 0.73 mm; Slightly greater pain on the 2nd day; Oedema similar to the control group, total reduction on the 4th day; Group 2: PRF Vestibular bone resorption: 1.26 ± 0.58 mm; Lingual bone resorption: 1.40 ± 0.66 mm; Horizontal bone gain: 1.46 ± 0.44 mm; There was no statistically significant difference in patient morbidity or crestal and horizontal bone alterations between trial groups.

Legend: β-TCP—β-Tricalcium Phosphate; PRP—Platelet Rich Plasma; PRF—Platelet Rich Fibrin; HA—Hydroxyapatite; A-PRF—Advanced Platelet Rich Fibrin; BIC—Bone-Implant Contact; REC—Gingival Recession; BOP—Bleeding on probing; CS—Calcium Sulphate; IT—Insertion Torque; ASA—American Society of Anaesthesiologists Physical Status Classification; PB—Bacterial Plaque; CBCT—Cone Beam Computed Tomography.

## Data Availability

The original contributions presented in this study are included in the article. Further inquiries can be directed to the corresponding author.

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
