# Peer review of "The Use of Platelet-Rich Fibrin in Combination with Synthetic Bone Grafting: A Systematic Review"

_biomedicines, 2025, doi:10.3390/biomedicines13092266_

Round 1

Reviewer 1 Report

Comments and Suggestions for Authors

This article provided a systematic review on the role of PRF in bone regeneration in sub-antral surgery. In addition, the combined/beneficial effects of PRF with nanocrystalline hydroxyapatite, beta-TCP and CS  biomaterials following subantral surgery were also listed in Table 4.  The authors also gave a detailed description regarding the methodology for selecting relevant studies used for this review article as listed in Tables 1, 2 and 3. 
In general,  this manuscript is well written in terms of clinical aspects of  PRF and its combined effects with nHA, CS and beta-TCP. However,  it is somewhat lacking with  scientific discussion on the mechanisms responsible for the combined effects of PRF with nHA, beta-TCP and CS.  By adding few discussions on the above-mentioned mechanisms, this article would become perfect for  publishing in this journal. 

Author Response

Dear Reviewer

Firstly, we would like to thank you for reviewing our manuscript and for all the proposed changes, which we believe have been essential to improving our work.

Throughout the document you can find all our answers highlighted in green.

Reviewer 1

This article provided a systematic review on the role of PRF in bone regeneration in sub-antral surgery. In addition, the combined/beneficial effects of PRF with nanocrystalline hydroxyapatite, beta-TCP and CS  biomaterials following subantral surgery were also listed in Table 4.  The authors also gave a detailed description regarding the methodology for selecting relevant studies used for this review article as listed in Tables 1, 2 and 3. 
In general,  this manuscript is well written in terms of clinical aspects of  PRF and its combined effects with nHA, CS and beta-TCP. However,  it is somewhat lacking with  scientific discussion on the mechanisms responsible for the combined effects of PRF with nHA, beta-TCP and CS.  By adding few discussions on the above-mentioned mechanisms, this article would become perfect for  publishing in this journal. 

You are right, we have added this information in discussion section as suggested. It is highlighted in green. (Please see lines 273-280; 285-290)

Best Regards 

Reviewer 2 Report

Comments and Suggestions for Authors
  • The exact range of dates for published articles uses in this study must be highlighted in the Abstract and Results sections. Seems like the earliest publication utilized in this analysis was in 2012. It is unclear why the team did not consider going back to much earlier articles that could be eligible to be considered in this review.
  • Make sure all the abbreviations are well defined in the first time they appear (e.g., PICO’s strategy)
  • Would be helpful if some of the quantitative pieces of data could be presented in proper graphs (e.g., "Regarding the type of study of the articles analysed, 5 are randomised clinical trials (46%), 186 4 are cohort studies (36%), and 2 are case series studies (18%)")
  • Tables need to be further improved to provide complete and consistent level of details for each study. Example: For the "Canullo et al. (2012)" study, in Table 4, the Sample data seems incomplete: "N = 10 patients Gender: N/A Mean age: 54 years" - not clear whether NanoBone® or some other bone regen method was used there.

Author Response

Dear Reviewer

Firstly, we would like to thank you for reviewing our manuscript and for all the proposed changes, which we believe have been essential to improving our work.

Throughout the document you can find all our answers highlighted in pink.

Reviewer 2

  • The exact range of dates for published articles uses in this study must be highlighted in the Abstract and Results sections. Seems like the earliest publication utilized in this analysis was in 2012. It is unclear why the team did not consider going back to much earlier articles that could be eligible to be considered in this review.

The search strategy was carried out without specifying the year in order to obtain all articles available in the literature on the subject. Although this topic is not new, there are many older studies on the use of PRF + xenograft, but there are no older articles that relate PRF to synthetic bone. That is why the selected articles are not very old. The date of the article search is already described in the materials and methods section of the abstract, as suggested. It is highlighted in pink. Please see line 19.

  • Make sure all the abbreviations are well defined in the first time they appear (e.g., PICO’s strategy)

You are right, we have added this information as suggested. (Line 111).

  • Would be helpful if some of the quantitative pieces of data could be presented in proper graphs (e.g., "Regarding the type of study of the articles analysed, 5 are randomised clinical trials (46%), 186 4 are cohort studies (36%), and 2 are case series (18%)")

We have added this information as suggested. It is highlighted in pink. (Lines 189-192).

  • Tables need to be further improved to provide complete and consistent level of details for each study. Example: For the "Canullo et al. (2012)" study, in Table 4, the Sample data seems incomplete: "N = 10 patients Gender: N/A Mean age: 54 years" - not clear whether NanoBone® or some other bone regen method was used there.

You are completely right. By mistake we forgot to add this information. We have added the information as suggested. 

Best Regards 

Reviewer 3 Report

Comments and Suggestions for Authors

The manuscript presents a systematic review on the use of platelet-rich fibrin (PRF), either alone or in combination with synthetic bone substitutes, for enhancing bone formation, implant stability, and survival in sub-antral surgery. While the topic is not new, the focus on clinical trials involving PRF with synthetic bone substitutes remains valuable and worthy of further investigation. However, the manuscript suffers from issues of poor organization and clarity.

Major Comments:

1. Title – The current title does not clearly reflect the central focus on PRF. A revision highlighting this aspect is recommended.

2. Materials and Methods – This section contains excessive detail, and some content appears redundant. The presentation should be revised for conciseness, logical flow, and clarity. In addition, the section of statistical method is missing. 

3. Results – The results section appears to only include several tables that are poorly organized, making it difficult for readers to grasp the key findings. In addition to the tables, the summarized narrative in paragraphs, as typically found in systematic reviews, should be added to highlight the most important outcomes.

Author Response

Dear Reviewer

Firstly, we would like to thank you for reviewing our manuscript and for all the proposed changes, which we believe have been essential to improving our work.

Throughout the document you can find all our answers highlighted in yellow.

Reviewer 3

The manuscript presents a systematic review on the use of platelet-rich fibrin (PRF), either alone or in combination with synthetic bone substitutes, for enhancing bone formation, implant stability, and survival in sub-antral surgery. While the topic is not new, the focus on clinical trials involving PRF with synthetic bone substitutes remains valuable and worthy of further investigation. However, the manuscript suffers from issues of poor organization and clarity.

Major Comments:

  1. Title – The current title does not clearly reflect the central focus on PRF. A revision highlighting this aspect is recommended.

You are right , we have changed the title as suggested. It is highlighted in yellow. (Lines 1-3)

  1. Materials and Methods – This section contains excessive detail, and some content appears redundant. The presentation should be revised for conciseness, logical flow, and clarity. In addition, the section of statistical method is missing. 

We did not include statistical analysis in this section because our systematic review did not use statistical evaluation.

  1. Results – The results section appears to only include several tables that are poorly organized, making it difficult for readers to grasp the key findings. In addition to the tables, the summarized narrative in paragraphs, as typically found in systematic reviews, should be added to highlight the most important outcomes.

You are right , we have reorganised the results as suggested.

Round 2

Reviewer 3 Report

Comments and Suggestions for Authors

The authors have addressed the comments. I have no further questions.